

# Compatibility of different measurement techniques. Long-term global solar radiation observations at Izaña Observatory

Rosa Delia García[1,2], Emilio Cuevas[1], Omaira Elena García[1], Ramon Ramos[1],
Pedro Miguel Romero-Campos[1], Fernado de Ory[1], Victoria Eugenia Cachorro[3], and Angel de Frutos[3]

[1]Izaña Atmospheric Research Center (IARC), State Meteorological Agency (AEMET), Spain
[2]Air Liquide España, Delegación Canarias, Candelaria, 38509, Spain
[3]Atmospheric Optics Group, Valladolid University, Valladolid, Spain

*Correspondence to:* R. D. García
(rgarciac@aemet.es)

**Abstract.** A 1-year intercomparison of classical and modern radiation and sunshine duration instruments has been performed at Izaña Atmospheric Observatory (IZO) located in Tenerife (Canary Islands, Spain) starting on July 17, 2014. We compare global solar radiation (GSR) records measured with a CM-21 pyranometer Kipp & Zonen, taken in the framework of the Baseline Surface Radiation Network, with those measured with a Multifilter Rotating Shadowband Radiometer (MFRSR), and a bimetallic pyranometer (PYR), and GSR estimated from sunshine duration performed by a Campbell-Stokes sunshine recorder (CS) and a Kipp & Zonen sunshine duration sensor (CSD). Given the GSR BSRN records are subject of strict quality controls (based on principles of physical limits and comparison with the LibRadtran model), they have been used as reference in the intercomparison study. We obtain an overall root mean square error (RMSE) of ∼0.9 MJm2 (4%) for GSR PYR and GSR MFRSR, 1.9 MJm2 (7%) and 1.2 MJm2 (5%) for GSR CS and GSR CSD, respectively. Factors such as temperature, fraction of the clear sky, relative humidity and the solar zenith angle have shown to moderately affect the GSR observations. As application of the methodology developed in this work, we have re-evaluated the GSR time series between 1977 and 1991 obtained with two PYRs at IZO. By comparing with coincident GSR estimates from SD observations, we probe the high consistency of those measurements and their temporal stability. These results demonstrate that 1) the continuous-basis intercomparison of different GSR techniques offers important diagnostics for identifying inconsistencies between GSR data records, and 2) the GSR measurements performed with classical and more simple instruments are consistent with more modern techniques and, thus, valid to recover GSR time series and complete worldwide distributed GSR data. The intercomparison and quality assessment of these different techniques have allowed to obtain a complete and consistent long-term global solar radiation series (1977-2015) at Izaña.

## 1 Introduction

The Earth's radiation budget is essential for driving the general circulation of the atmosphere and oceans, and modulating the main conditions of the Earth's climate system. Several studies have examined the evidence of links between recent changes in climate and in the amount of global solar radiation (GSR) reaching the Earth's surface, observing a decrease in the



GSR at the surface between the 1960s and the 1990s, an effect known as dimming, with a general decline between 4% and 6% decade$^{-1}$ over 30 years considering worldwide distributed stations (e.g. Ohmura and Lang (1989); Gilgen et al. (1998); Stanhill and Cohen (2001); Wild et al. (2005); Wild (2009)). In contrast, an increase of the GSR between 1% and 10.7% decade$^{-1}$ since the 1980s has been documented (e.g. Wild et al. (2005, 2007, 2008); Wild (2012); Gilgen et al. (2009)), known as brightening. However, for a better understanding of the global effects in the climate system, long-term GSR time series in representative regions are fundamental.

The first solar radiation instruments were designed in the first decade of the last century (Moll, 1913; Chaldecott, 1954; De Bruin et al., 1995; Stanhill, 1998), however, continuous GSR observations began in the 1920s at selected locations. The longest known GSR time series have been measured in Stockholm (Sweden) since 1923 (Stanhill and Cohen, 2001), Wageningen (Netherlands) since 1928 (De Bruin et al., 1995) and Potsdam (Germany) since 1937 (Wild, 2015). However, regular and coordinated GSR measurements were not well established until the framework of the International Geophysical Year in 1957 and 1958 (IGY, 1957/1958) (Nicolet, 1982). The efforts made to increase the knowledge and measurement of the GSR lead to the necessity to centrally collect the GSR measurements. In 1964, the World Meteorological Organization (WMO) established the World Radiation Data Centre (WRDC), which has been operated for over 50 years supported by the Main Geophysical Observatory of the Russian Federal Service for Hydrometeorology and Environmental Monitoring, and centrally collects archives and published radiometric data from several National Meteorological and Hydrometeorological Services and other organizations (WMO, 1965). Furthermore, the increase in the comprehension of the GSR influence in the climate system required a homogenization in the accuracy of the measurements. Thus, several GSR measurement networks were established around the world in the early 1990s. In 1992 the Baseline Surface Radiation Network (BSRN; http://www.bsrn.awi.de) was created (Heimo et al., 1993; Ohmura et al., 1998). The BSRN is characterized for meeting strict quality control and quality assurance protocols, and consists nowadays of approximately 60 stations in diverse climatic regions across the globe, whose radiation measurements are widely used in climate models and satellite calibration algorithms (Heimo et al., 1993; Ohmura et al., 1998). Other examples are the Atmospheric Radiation Program (ARM, Ackerman and Stokes (2003)), set up in 1992 or the Surface Radiation Budget Network (SURFRAD, Augustine et al. (2000)) created in 1993.

Unfortunately, before the establishment of the cited networks, long-term GSR time series were very scarce and not representative for a wide variety of atmospheric conditions. In addition, throughout history, instruments of different types have been used for measuring GSR or to obtain indirect estimations of GSR, as sunshine duration (SD), with different accuracies, and even different radiometric scales: All this has caused inconsistencies and inhomogeneities in radiation data series, mainly before the 1950s (Fröhlich, 1991). On the other hand, new instruments with better accuracies have regularly replaced older ones, resulting in new uncertainties since in many instrument replacements there were no simultaneous measurements allowing to know the compatibility between old and new instruments and the corresponding changes caused in data series.

In this context, this work compares ground-based GSR measurements performed by different instruments with GSR derived from SD measurements in order to 1) document the traceability of main solar radiation techniques historically used, and thereby 2) assess their suitability for completing and recovering GSR time series, valid for climate studies. The Izaña Atmospheric Observatory (IZO) is an optimal station to carry out this quality assessment study, since solar radiation observations have been



continuously performed since the early 1920s. At IZO, the SD observations started in 1917 with a Campbell-Stokes sunshine recorder (onwards, CS recorder), which was recently replaced by a Kipp & Zonen sunshine duration sensor (onwards, CSD recorder). The GSR measurements started in 1977 with a bimetallic pyranometer (PYR) and it was replaced in 1992 by different instruments (Kipp & Zonen: CM-5, CM-11 and CM-21). Since 2005, IZO has been a station member of BRN managed by

the Spanish National Radiometric Centre (NRC-AEMET) and since 2009 IZO has been part of the BSRN (BSRN station #61, IZA) (García et al., 2012, 2014c). This work is divided in six sections. Section 2 describes the main characteristics of IZO test site. Section 3 shows the technical description of ground-based instruments and the methodology used to derive the GSR estimations from the different observations. Section 4 presents the results obtained in a 1-year intensive campaign ad-hoc designed to compare different GSR measurement techniques, and finally, in Section 5 the GSR time series performed with two

pyranometers between 1977 and 1991 at IZO is evaluated. A summary and the main conclusions of this work are given in Section 6.

## 2   Site Description

The Izaña Atmospheric Observatory (IZO, http://izana.aemet.es) is located on the island of Tenerife (Canary Islands, Spain; at 28.3°N, 16.5°W, 2373 m a.s.l.), and it is managed by the Izaña Atmospheric Research Center (IARC), which forms part of the

Meteorological State Agency of Spain (AEMET).

IZO is a high-mountain observatory above a strong subtropical temperature inversion layer, which acts as a natural barrier for local pollution, and it provides atmospheric measurements representative of free troposphere conditions of the North Atlantic region due to the quasi-permanent subsidence regime typical of the subtropical region (Cuevas et al., 2013; Gómez-Pelaez et al., 2013). It is optimal for solar radiation measurements given that it has about 3436 hours of sunshine per year and a mean annual

number of cloud-free days of 221 (61%) between 1916 and 2015, as well as for calibration and validation activities due to a high atmospheric stability, stable ozone total column amounts, very low column water content and low aerosol concentrations. Due to these privileged measurement conditions IZO has developed a comprehensive atmospheric monitoring program. In 1984 IZO became a member of the World Meteorological Organization (WMO) Background Atmospheric Pollution Monitoring Network (BAPMoN) and in 1989 it became a Global Atmosphere Watch (GAW) station. Also, it has actively contributed to interna-

tional radiation networks and databases such as the Network for the Detection of Atmospheric Composite Change (NDACC, http://www.ndsc.ncep.noaa.gov) and the Precision Filter Radiometer Network (GAW/PFR, http://www.pmodwrc.ch/worcc) since 2001, the Aerosol Robotic Network (AERONET, http://aeronet.gsfc.nasa.gov) since 2004, and the BSRN since 2009, among others. Moreover, in July 2014, IZO was appointed by WMO as a CIMO (Commission for Instruments and Methods of Observation) Testbed for aerosols and water vapour remote sensing instruments (WMO, 2014).



## 3 SOLAR RADIATION INSTRUMENTS AND METHODOLOGY

In order to compare different GSR measurement techniques, all the solar radiation instruments that have historically measured solar radiation at IZO were re-installed at the observatory performing simultaneous observations between July, 17 2014 and July, 12 2015. The installed instruments were a Robitzsch bimetallic pyranometer (also known as pyranograph or actinograph,

hereafter PYR), a Campbell-Stokes sunshine recorder (CS recorder), a Kipp & Zonen sunshine duration sensor (CSD recorder), a Multifilter Rotating Shadowband Radiometer (MFRSR) and a CM-21 pyranometer Kipp & Zonen (hereafter BSRN) (see Table 1). A CM-21 Kipp & Zonen pyranometer from the BSRN program was used as reference. In the following sections the different instruments are described as well as the methodology used to derive the GSR.

### 3.1 BSRN station

The GSR measurements taken as reference in this study are those from the Izaña BSRN. Since 2009, IZO has been part of the BSRN (#61, IZA) (García et al., 2012); http://www.aemet.bsrn.es). The GSR at Izaña BSRN is measured with a Kipp & Zonen CM-21 pyranometer (Table 1). The pyranometer performs both diffuse and global solar radiation measurements. The pyranometers integrate radiation hemispherically over a horizontal receiver surface and cover a spectral range from 310 to 2800 nm (95% points). The measurements are sampled at 1 Hz and 1-min mean values are recorded. This instrument has been

calibrated recently at the World Radiation Center (WRC) at Davos, Switzerland and it is regularly compared at IZO with a PMO6 absolute open cavity radiometer (reference instrument) designed at the Physikalisch-Meteorologisches Observatorium Davos(PMOD) following the ISO 9059:1990 (E) and ISO 9846:1993 obtaining less than 0.1% difference between the given calibration coefficient obtained in WRC and IZO.

The expected uncertainty is $\pm 2\%$ for instantaneous, hourly and daily totals (Ohmura et al., 1998; McArthur, 2005). Similar

values were found by García et al. (2014c), they present a comparative study of GSR measurements and simulations at BSRN Izaña station. The analysis was based on cloud-free days between March 2009 and August 2012 (386 days), including both aerosol-free and Saharan almost-pure mineral dust conditions. They observed agreement within 99% and the bias (simulations-measurements) was -0.30$\pm$0.24 MJm$^{-2}$(-1.1$\pm$0.9%) and RMSE of 0.38 MJm$^{-2}$ (1.3%). However, Stoffel (2005) estimated larger uncertainties for field GSR measurements, $\pm 6\%$ (10 Wm$^{-2}$). These uncertainties are associated with radiometer calibra-

tion and measurement system installation, operation and maintenance. The BSRN establishes strict quality controls for shortwave surface radiation and other measurements (Long and Dutton, 2002; Long and Shi, 2006). Applying the BSRN quality controls aforementioned to the Izaña GSR measurements for solar zenith angles (SZA)<90°, García et al. (2014c) shows that the measurements largely satisfy the quality control recommended by the BSRN obtaining less than 0.1% of the measurements out of the physically possible and extremely rare limits (see Table 1 of García et al. (2014c)). Also, other validation of the GSR

measurements is routinely made by comparison with the LibRadtran model (http://www.libradtran.org; Mayer and Kylling (2005)) reporting a high consistency between simulations and measurements of GSR (see Table 5 in García et al. (2014c)).





### 3.2 Bimetallic pyranometer

The PYR was designed by Max Robitzsch in 1926 (Table 1), and it was a popular instrument for measuring daily amount of solar irradiance from about 1932 to 1970 at meteorological stations around the world, since it is easy to install, to operate, and has low-maintenance. The pyranometer is based on the properties of a bimetallic strip mounted under a protective glass
dome, which bends in response to temperature changes. The different thermal expansion properties of the materials used to form the bimetallic strip create a simple bimetallic thermometer that responds proportionally to changes in solar irradiance (Vignola et al., 2012). By coating the strip with a black absorbent material and exposing it to the sun's energy, the strip bends in proportion to the change in temperature, which is proportional to the intensity of the solar radiation. The signal is instantaneously recorded in a strip card, being the area under the radiation curve proportional to the incoming solar radiation. The
measurement uncertainties introduced by a variety of undesirable response characteristics result in daily total irradiance within $\pm 20\%$ for the best performing instruments (Coulson, 1975; Garg and Garg, 1993). For this reason this instrument was classified by WMO as a third-class pyranometer (WMO, 1965).

In order to convert the area obtained from the pyranometer strip card to radiation units the following equation is used:

$$Q = K \ S \ (1 + 0.0033 \ \bar{t}) \tag{1}$$

where $K$ is the instrument constant, $S$ is the value of the integrated surface (area) in cm$^2$ and $\bar{t}$ is the averaged temperature
in °C form temperature observations at 07-13-18 UTC. $Q$ is expressed in cal cm$^{-2}$ min$^{-1}$. The constant $K$ is dependent on each instrument; unfortunately this constant is unknown for the instrument used in this work. Therefore, we have estimated the constant $K$ from equation 1 considering that $Q$ is the theoretical GSR. In particular, we have calculated a $K$ value for each month considering only cloud-free days. These days were selected using the Long and Ackerman's method (Long and Ackerman, 2000). Hereafter, the GSR estimations with this method are denoted GSR PYR-1.

Also, we have used a second approach, the Esteves's method (Esteves and de Rosa, 1989), to determine the GSR performed with the pyranometer (hereafter, GSR PYR-2). This method defines a monthly factor ($F_m$) that is determined for a given set of cloud-free days of the same month as the sum of ratios between the theoretically calculated daily GSR value and the area of the register for the same day, divided by the number of days in the set:

$$F_m = \frac{\sum_{i=1}^{n_m} \frac{H_{ci}}{S_i}}{n_m} \tag{2}$$

where $H_{ci}$ is the theoretical GSR for the clear day $i$, $S_i$ is the area inscribed under the registered curve for the day i and $n_m$ is
the number of clear days of the month considered. Once the value of the monthly factor is obtained, it is possible to calculate the daily GSR for any day of the month as follows:

$$GSR = F_m \ S \tag{3}$$

where $S$ is the area inscribed under the register's curve for a given day of the month (idem to equation 1). GSR is expressed in MJm$^{-2}$.



For the two methods, two variables are needed to be calculated: the theoretical GSR ($Q$ in PYR-1 and $H_{ci}$ in PYR-2) and the area $S$. Firstly, the theoretical GSR was calculated using the LibRadtran model (see Section 3.1). To simulate the GSR values, we used daily aerosol optical depth (AOD) data from AERONET (Holben et al., 1998) and total precipitable water vapour (PWV) observations from Vaisala meteorological radiosondes (Miloshevich et al., 2009), launched at Güimar (WMO station

#60018) at 11:15 UTC (Romero Campos et al., 2011). Refer to (García et al., 2014c) for more details about the LibRadtran simulations.

The area, S, was calculated by using a semiautomatic method ad-hoc developed at IZO to digitalize the data recorded on the pyranometer strip cards. Figure 1 shows the flow chart illustrating the methodology used on August 10, 2014, as an example. The processing is performed using the Matlab Image Processing Toolbox (Russ, 1994). Firstly, each strip card is scanned

with a resolution of 300 ppi and saved to 24-bit RGB TIF (Tagged Image Format). The image is trimmed in order to only include the radiation curve in the image, discarding possible spurious notations and the signals on the strip card. In a few cases, minor editing of the trimmed image is needed to clean it of spots and defects. A median filter is applied to the trimmed image and two background areas are selected (Figure 1(3)), normally in the upper corners of the image. For each channel (R, G and B) a statistical analysis is performed on the background areas in order to stablish the thresholds needed to define the

pixels that belongs to the curve. Once the curve pixels are identified we extract its coordinates in pixel units and the curve is smoothed to detect the sunrise and sunset points using the Savitzky-Golay algorithm (Gorry, 1990) (Figure 1(6)). With the pixel coordinates of the sunrise and sunset we construct a straight line connecting both points. Finally the area, S, in pixels$^2$ is obtained by calculating the area bounded between the radiation curve and the base line. Taking into account that the resolution of the image is 300 ppi, dividing the area S in pixels$^2$ by 13950, the area in cm$^2$ is obtained.

We have also computed the instantaneous GSR values from the pyranometer radiation curve. The conversion of the horizontal axis from pixels to time is straightforward by taking in mind the relation between the distances in pixels and time from the sunrise to the sunset throughout the day. The vertical conversion requires more effort due to the clearly dependence of the calibration factors with month and hour, as pointed by previous works (Albrecht, 1954; Stravisi, 1986). Thus, calibration factors dependence on the mentioned variables is required. In order to convert from cm to Wm$^{-2}$ the next relation is proposed:

$$GSR \ PYR \ (Wm^{-2}) = C \ (month, minute) \ Y \ (mm) \tag{4}$$

where Y is obtained in the image processing method already discussed and C are coefficients computed with the LibRadtran model which has demonstrated to be a very useful measurement quality control tool (García et al., 2014c, a). This study is performed for a sample of cloud-free days in May and June (2015), a month in which there is a maximum SZA range.

## 3.3 Sunshine recorders: CS and CSD

The CS recorder was invented by J. F. Campbell in 1853 and modified to its usual shape by Sir G. G. Stokes in 1879 (Table 1).
The CS focuses the direct solar beam through a glass sphere, mounted concentrically in a section of a spherical bowl, on a burn card located under the sphere. The card is provided with a time indication, which makes it possible to determine the sunshine duration (SD) from the length of the burn when the card is removed from the instrument at the end of the day (Painter, 1981;

WMO, 1996; Sanchez-Lorenzo et al., 2013; García et al., 2014a). The errors of this recorder are mainly due to the dependence of burning initiation on card's temperature and humidity as well as to the overburning effect, especially in case of broken clouds (Kerr and Tabony, 2004).

The CSD recorder (Table 1) was designed with the objective of automating the process of measuring SD for meteorological services and has replaced the traditional CS recorders. This instrument is formed by three detectors: the sensor at the front measures global radiation, while the other two detectors, at the middle and at the rear, are partially shaded covering 1/3 of the sky, measuring diffuse radiation (Kipp and Zonen, 2003). Furthermore, the three detectors have exactly the same spectral and angular characteristics, making the process of calibration easy. The direct solar radiation can be computed from the values of global and diffuse solar radiation, and the sunshine duration is determined according to the latest WMO definition (WMO, 1996).

In this work, the method used to determine the GSR estimated from SD records was developed by Ångström (Angstrom, 1924, 1956) and later modified by Prescott and Rietveld (Prescott, 1940) and applies the following equation:

$$\frac{H}{H_o} = a\frac{n}{N_d} + b \tag{5}$$

where $H$ and $Ho$ are the daily GSR and the daily extraterrestrial GSR ($MJm^{-2}day^{-1}$), respectively, on a horizontal surface, $n$ and $Nd$ are the number of hours measured by the SD recorder and the maximum daily SD, respectively, and $a$ and $b$ are coefficients to be determined by using linear regression.

This method has successfully been validated at IZO by García et al. (2014a, b), obtaining an expected uncertainty of 9.2% for CS recorder and 5.5% for CSD recorder when comparing BSRN GSR. Following equation 5 and details given in García et al. (2014a, b) we have calculated the a and b coefficients at IZO are given in Table 2 for each SD recorder. To account for the variability introduced by the presence of clouds, fog, etc., these coefficients were estimated in function of the fraction of clear sky (FCS), which is defined as the ratio between the maximum daily sunshine duration $Nd$ and SD performed with CS recorder ($n$):

$$FCS(\%) = \frac{n}{N_d} * 100 \tag{6}$$

### 3.4 MFRSR radiometer

The MFRSR (Table 1) was developed in the early 1990s. This radiometer uses independent interference-filter-photodiode detectors and the automated rotating shadow-band technique to make spectrally resolved measurements at seven-channels with six wavelength passbands of 10 nm FWHM centered near 415, 500, 610, 665, 862, and 940 nm, and an unfiltered channel used to obtain broadband solar irradiance estimates from 300 to 1100 nm. Each measurement sequence is routinely repeated every 15 s and the recorded instantaneous signals are subsequently averaged over a 1-min time interval to increase the signal-to-noise ratio (Harrison et al., 1994).

In this work, the MFRSR was re-calibrated using the LibRadtran model obtaining a new calibration factor calculated as the mean of ratio between the GSR measurements and simulations in a time interval of 5 minutes before and after the solar noon (WMO, 1996), only considering cloud-free days (160 days, 44%). The obtained calibration factor was 0.438±0.008.





Although the MFRSR provides the GSR directly, the spectral range covered by this instrument (300-1100 nm) is significantly smaller than that of the reference instrument used, the CM-21 pyranometer (280-2600 nm). For this reason, we have calculated a radiation correction using the LibRadtran model as a linear relation between the GSR in the MFRSR and the CM-21 spectral ranges. We have modelled the same days selected to calculate the calibration factor in both spectral ranges with the same input values, obtaining a very good agreement (correlation coefficient, R, of 0.995) and the following fit equation:

$$GSR_{280-2600} = (1.239 \pm 0.006)GSR_{300-1100} + (0.4247 \pm 0.014) \tag{7}$$

## 4 RESULTS

In this section we present the comparison of the daily GSR values obtained with the different instruments and techniques using BSRN as reference and we also perform an analysis of the GSR bias in function of the season, solar irradiation, temperature, relative humidity and aerosol optical depth (AOD). In addition, we present the comparison of the instantaneous GSR values measured with PYR and MFRSR with respect to BSRN.

### 4.1 Comparison of daily GSR data

Applying the different methodologies discussed in the previous section, we have computed the daily GSR values from the different instruments and techniques between July, 17 2014 and July, 12 2015 at IZO. The daily GSR was calculated according to following equation:

$$GSR = \int_{sr}^{ss} I(t)dt \tag{8}$$

where I(t) are the instantaneous values of the solar irradiance (Wm$^{-2}$) at time (t), computed from sunrise (sr) to sunset (ss). Figure 2 summarizes the straightforward comparison among them.

In general, and expected, the best agreements with GSR BSRN (with R=0.999) are found for those instruments that directly measure solar radiation (nor estimations) and have been calibrated with the LibRadtran model, since the model has been widely validated against the BSRN measurements at IZO (García et al., 2014b). Although the poorest scores are those provided by GSR estimations from CS measurements (R=0.978), it is worth to emphasize that the correlation coefficient is always greater than 0.97 and the slope ranges between 0.92 and 1.04 for all cases. Regarding the correlation between GSR PYR-1 and GSR PYR-2 is very good with R= 0.997.

In order to quantify the uncertainty of the different techniques with respect to GSR BSRN, we have calculated the absolute difference or bias (GSR XXX – GSR BSRN, in MJm$^{-2}$) and the relative difference ((GSR XXX – GSR BSRN)/GSR BSRN, in %), where XXX means PYR, CS, CSD and MFRSR. As a summary, Table 3 lists the metrics used to quantify these measurements (MB: median bias; STD: standard deviation and RMSE: root mean square error). In general, GSR is underestimated, except for GSR MFRSR. The results obtained for GSR PYR-1 and PYR-2 are very similar with MB of -0.5 MJm$^{-2}$ (-2%), while the precision, given by the RMSE, is of 0.9 MJm$^{-2}$ (3.5%) for both cases. The greatest difference is obtained for the



GSR CS, with MB of -1.14 MJm$^{-2}$ (-4.5%) and RMSE of 1.70 MJm$^{-2}$ (6.7%). The GSR MFRSR is slightly overestimated with a MB of 0.28 MJm$^{-2}$ (1.1%) and RMSE of 0.54 MJm$^{-2}$ (2.1%). Since the results obtained by PYR 1 and PYR-2 are similar, hereafter we will discuss only the results of the PYR-1 and rename as PYR.

As aforementioned, some of instruments and method analyzed are sensitive to different factors or atmospheric conditions.
To analyze the impact of these factors on the inter-comparison analysis, we have studied the differences with respect to the BSRN in function of the season (Figure 3), the solar irradiance (measured GSR BSRN; Figure 4a, the average temperature, the averaged relative humidity (RH) from observations at 07, 13, and 18 UTC at IZO (Figure 4b and 5 4c, respectively), and the aerosol optical depth (AOD; Figure 5). A seasonal variation is observed for all GSR measurements. In general GSR is underestimated for all seasons and instruments except for GSR MFRSR in winter (DJF) and autumn (SON). The higher
scatters are found in the GSR PYR and GSR CS while the lower are found in the GSR MFRSR for all seasons.

The GSR PYR estimations present higher median bias in winter and spring than in summer and autumn, while the scatter are higher in spring and summer. There is not dependence with solar irradiance level (Figure 3a), with a median bias almost constant around -0.5 MJm$^{-2}$. The GSR PYR shows a slight temperature dependence with the lowest median bias in the 10-20°C range (Figure 4b) that agrees with the seasonal variation, considering that the highest median temperatures are found in
summer and autumn at IZO (Figure 3a). These results also agree with previous works (Stravisi, 1986). No clear dependence with relative humidity (Figure 3c) is found. This behavior may be attributed to the response of the bimetallic strips to temperature changes (see Section 3.2), since the optimum operating temperature is around 15°C, explaining the lower bias values in the warmer seasons (Figure 3a and c).

The GSR CS estimations show the highest median bias in spring and summer, and slightly lower in autumn and winter
(Figure 3). It is clear a dependence with the irradiance level, the larger BSRN GSR values the larger bias. However there is not dependence with temperature with an almost constant bias value around -1 MJ$^{-2}$ (Figure 4b). A slightly dependence with the relative humidity, being the highest bias found for the lowest RH values (Figure 4c) is noticed This might be explained by the fact that the card requires more solar irradiance to be burn under wetter conditions, as frequently occurs in early morning (Bider, 1958), while it is easier to burn it under warm and dry conditions (Kerr and Tabony, 2004; Painter, 1981). Since lower
values of relative humidity and higher temperatures facilitate the burning of the card, an over burning of the card in spring and summer seasons is expected in comparison with winter.

The GSR CSD estimations, unlike GSR CS, present much lower bias, showing an opposite annual cycle to that found for GSR CS bias, with higher GSR bias in winter and autumn, and lower in spring and summer. Although a slight dependence on irradiance is observed, this is much lower than in case of GSR CS (Figure 4a). There is a slight dependence with temperature,
with higher negative bias for lower temperatures (Figure 4b) and a median bias close to zero for temperatures >20°C. The manufacturer declares a temperature dependence of 0.1%/°C (Kipp and Zonen, 2003). There is no dependence with relative humidity (Figure 4c).

Finally the MFRSR is the instrument that provides the best performances, with bias close to zero through the whole year and the lowest scatter (Figure 3). We observe, in general, an overestimation in MFRSR GSR, unlike the rest of the compared
instruments. The GSR MFRSR has a clear dependence with irradiance, increasing the bias with the latter (Figure 4a). There



is not temperature dependence for temperatures >15°C, (Figure 4c) and slight dependence, with lower bias for lower temperatures. The MFRSR is thermally controlled at 40°C. The GSR MFRSR measurements show a slight dependence with relative humidity, increasing the bias with the latter (Figure 4c). These results agree with the seasonal dependence which shows the best results in spring and summer. We have also studied the differences with respect to FCS (no shown here), but they do not

have statistical significance because 85% of the days in the sample (N:232 days) present FCS>75% and only the 1% (N:4 days) present FCS<25%. Anyway no dependence with FCS was found.

However, the most significant GSR bias dependence of all instruments, except the PYR, is with the column aerosols content (AOD; see Figure 5). PYR practically does not show any dependence. On the contrary, CS and MFRSR show a strong increase in GSR bias only for very high AOD (i.e., >0.6), and the CSD shows a monotonic dependence in GSR bias, being negative

(high in absolute value) for pristine skies, and positive (high in absolute value) for strong dusty skies. Aerosols content reveals as the main factor driving the GSR bias in all instruments except the PYR. Simulations performed with the LibRadtran model on broad range of AOD values for clear skies, using actual input parameters, indicate no significant dependence of GSR BSRN with AOD (not shown here).

## 4.2 Diurnal GSR comparison

This section presents the diurnal comparison of the instantaneous GSR values from PYR and MFRSR (the only instruments that have this information) with those of the BSRN, used again as reference, applying the methodology discussed in section 3.2. For that, we have considered 27 cloud-free days between May and June 2015, grouping them into intervals of SZA from 10° to 60° (Table 4, Figure 6). The results show that the GSR MB decreases with the SZA for PYR GSR, while MB is quite stable for MFRSR GSR values (Table 4 and Figure 6). In both cases the STD increases notably with SZA.

## 5  1977-1991 TIME SERIES

As an application of the previous analysis, we have applied the methodology explained in Section 4.1 to the measurements performed at IZO between 1977 and 1991 with two pyranometers: 1) #290604 (1977-1982), and 2) #250585 (1983-1991). The theoretical GSR needed to obtain the instrument constant ($K$ in equation 1, GSR PYR-1) in these periods was calculated using the LibRadtran model, where the main model inputs were the AOD and PWV. Given that there are no available AOD

measurements, we have used AOD derived from artificial neural networks (ANNs) applying the methodology developed by García et al. (2016). The PWV between 1977 and 1979 was taken from the NCEP/NCAR reanalysis (Kalnay et al., 1996) and since 1980 we have used the PWV obtained from meteorological radiosondes (Miloshevich et al., 2009), and specifically those launched from Santa Cruz de Tenerife station (WMO#60020). Once the instrument constants were evaluated we calculated the GSR from the PYR measurements. By combining the reevaluated GSR time series between 1977 and 1991 from PYR

measurements and the GSR series measured with different pyranometers from 1992 to 2015, obtained in García et al. (2014b), we finally completed the 38-year GSR time series at IZO, depicted in Figure 7.



In order to analyse the suitability of the PYR measurements to fill gaps in long-term GSR time series, we have studied the temporal stability of the PYR time series at IZO between 1977 and 1991 by comparing two independent GSR data series. During that period, the only available GSR data at IZO are those estimations from CS records, presented in García et al. (2014b). That work presented the re-evaluation of the GSR time series between 1933 and 1991 from SD measurements performed with

a CS by using the Ångström-Prescott method, and documented also its high quality and temporal consistency. Hereafter, we refer to that time series like GSR CS-AP to distinguish it from the GSR CS obtained in the current study.

In general, there is a good agreement between GSR PYR and GSR CS-AP (Figure 8a), being both GSR time series very consistent each other with correlation coefficient of 0.86. To check the temporal stability of GSR PYR observations, we examine possible drifts and discontinuities in the median bias time series (GSR PYR – GSR CS-AP), considering a drift as the linear

trend of the annual median bias, while the change-points (change in the median of the bias time series) are analyzed by using a robust rank order change-point test (Lanzante, 1996). We found that there are no significant drifts in the median bias time series and no change-points are found at 99% of confidence level. To complete this analysis, we have examined the temporal evolution of the annual transfer function between GSR PYR and GSR CS-AP, shown in Figure 8b. This is calculated as the annual slope and intercept considering all the GSR PYR and GSR CS-AP observations for each year. As for MB, neither temporal drifts nor

significant discontinuities were detected at 99% of confidence level for any of the least-square fit parameters.

The slope values are higher than 0.88 for all the years, being the median value of 0.92, coincident with that obtained between GSR PYR and GSR CS for the 2014-2015 period (Figure 3). However, the intercept is significantly higher in the 1977-1991 period than in the 2014-2015 period that might be likely due to, after reinstallation a correction in the bias, to instrument cleaning and fit. The goodness of the fit is noticeable during the whole period, with R values of about 0.95, whilst the obtained

between July, 17 2014 and July, 12 2015 is of 0.98 (see Figure 2).

## 6   SUMMARY AND CONCLUSIONS

A 1-long-year comparison (July, 17 2014 to July, 12 2015) between GSR measurements performed with old and modern radiation and sunshine duration instruments has been performed at IZO. The daily GSR values measured with a bimetallic pyranometer (PYR) and Multifilter Rotating Shadowband Radiometer (MFRSR) and GSR estimated from SD performed by

a Campbell-Stokes sunshine recorder (CS) and a sunshine duration sensor (CSD) have been compared with respect to GSR from a BSRN CM21 pyranometer. We have also compared instantaneous values of GSR performed with PYR and MFRSR for different SZA with respect to GSR BSRN.

Assuming GSR from BSRN as reference, the measured or estimated GSR values show median biases of 2% and 1% for GSR PYR and GSR MFRSR, respectively, and of 5% and 2% for GSR CS and GCS CSD, respectively. These results, as expected,

show that the instruments that measure directly GSR, such as the PYR and MFRSR, present lower MB and lower scatter than the ones that estimate the GSR, such as the CS and CSD recorders. Moreover, median MB values for each instrument are within their corresponding uncertainty, agreeing results reported by other authors (Coulson, 1975; García et al., 2014c; McArthur, 2005). Indeed the biases for PYR found in this study are lower than those reported by others authors. For example,



Coulson (1975); Garg and Garg (1993) obtained uncertainties between 10 and 20%. The comparison of the daily GSR values from PYR and MFRSR showed a good agreement with GSR BSRN, with a RMSE of ~0.9 MJm$^{-2}$ (3%) and ~0.5 MJm$^{-2}$ (2%) for GSR PYR and GSR MFRSR, respectively, and ~1.7 MJm$^{-2}$ (7%) and ~1.1 MJm$^{-2}$ (4%) for GSR CS and GSR CSD, respectively. These comparisons showed a slight dependence of GSR bias with temperature and relative humidity, that

varies for each instrument. However, the AOD is the parameter with which we have found the most relevant dependence, except for the PYR. The GSR bias from CS and MFRSR instruments show a strong increase for very high AOD (>0.6), and the CSD shows a monotonic dependence in GSR bias, being negative (high in absolute value) for pristine skies, and positive (high in absolute value) for strong dusty skies.

The GSR from PYR and MFRSR were compared with BSRN GSR on a daily basis at different solar zenith angles. The

RMSE increases from 3% to 7%, and from 0% to 3% for GSR PYR and GSR MFRSR, respectively, when the SZA increases.

The methodology developed in this work has allowed us to obtain a unique re-evaluated and quality assured GSR time series from 1977 to 1991 at IZO using GSR data from two bimetallic pyranometers. The consistence in the obtained daily GSR PYR is good with respect to the daily GSR estimates from SD measurements with a correlation coefficient of 0.86.

These results demonstrate that: 1) The continuous intercomparison of different techniques used to measure GSR constitutes

an important diagnostics tool for identifying inconsistencies between GSR data records, and 2) GSR measurements performed with classical and simple instruments are fundamental for filling gaps in long-term GSR data series after accurate data screening, calibration and correction by using ad-hoc reanalysis techniques based in transference radiative models.

*Acknowledgements.* This is study forms part of the activities carried out within WMO CIMO Testbed for Aerosols and Water Vapour Remote Sensing instruments at Izaña Observatory. The authors are grateful to the IZO team and especially all observers who have worked in the past

70 years performing pyranograph bimetallic measurements at IZO as well as all members of security staff who have helped us to make possible this work.



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

**Table 1.** Characteristics and uncertainties of each solar radiation instrument used: Robitzsch bimetallic pyranometer (PYR), Campbell-Stokes sunshine recorder (CS recorder), Kipp & Zonen sunshine duration sensor (CSD recorder), Multifilter Rotating Shadowband Radiometer (MFRSR) and pyranometer CM-21 Kipp & Zonen (BSRN) installed between July, 17 2014 and July, 12 2015 at IZO.

| | *Pyranograph bimetallic* | *CS recorder* | *CSD recorder* | *MFRSR radiometer* | *CM-21 Kipp & Zonen Pyranometer* |
|---|---|---|---|---|---|
| *Instruments pictures at IZO* | | | | | |
| *Magnitude* | Area(cm$^2$) | SD(hours) | SD(hours) | GSR(MJm$^{-2}$) (300-1100 nm) | GSR(MJm$^{-2}$) (280-2600nm) |
| *Uncertainty* | ±5 to 10% | ±9% | ±5% | <5% | ≈2% |
| *Reference* | Garg and Garg (1993) | García et al. (2014b) | García et al. (2014a) | Hodges and Michalsky (2011) | Ohmura et al. (1998) |

**Table 2.** The *a* and *b* coefficients obtained between 1992 and 2000 for the CS recorder and between 2001 and 2013 for the CSD recorder, in function of FCS (fraction of clear sky, %) and % of days used (see (García et al., 2014b)). SEM is standard error of the mean.

| *FCS(%)* | *a±SEM* | *b±SEM* | *% days* |
|---|---|---|---|
| | *CS: 1992/2000* | | |
| ≤20 | 0.304±0.120 | 0.347±0.012 | 9 |
| 20-40 | 0.449±0.144 | 0.348±0.050 | 5 |
| 40-60 | 0.516±0.085 | 0.325±0.048 | 8 |
| 60-80 | 0.402±0.041 | 0.399±0.033 | 23 |
| ≥80 | 0.475±0.039 | 0.339±0.038 | 55 |
| | *CSD: 2001/2013* | | |
| ≤20 | 0.642±0.140 | 0.281±0.015 | 3 |
| 20-40 | 0.664±0.111 | 0.258±0.038 | 5 |
| 40-60 | 0.577±0.074 | 0.273±0.042 | 5 |
| 60-80 | 0.447±0.044 | 0.362±0.036 | 11 |
| ≥80 | 0.719±0.016 | 0.109±0.016 | 76 |

**Table 3.** Statistics of the comparison between daily GSR from PYR (Method 1 and 2), CS recorder, CSD recorder and MFRSR with daily GSR from BSRN (GSR XXX – GSR BSRN) at IZO between July, 17 2014 and July, 12 2015 (in MJm$^{-2}$). MB, median bias; STD, standard deviation; MAB, mean absolute bias and RMSE, root mean square error. The corresponding statistics in percentage are in brackets (%)

| | *MB* | *STD* | *RMSE* | *MAB* |
|---|---|---|---|---|
| *GSR PYR-1/BSRN* | -0.50 (-1.9%) | 0.73 (4.3%) | 0.88 (3.5%) | 0.74 (2.9%) |
| *GSR PYR-2/BSRN* | -0.54 (-2.2%) | 0.66 (4.5%) | 0.86 (3.4%) | 0.70 (2.8%) |
| *GSR CS/BSRN* | -1.15 (-4.4%) | 1.46 (6.4%) | 1.86 (7.1%) | 1.58 (6.2%) |
| *GSR CSD/BSRN* | -0.41 (-1.5%) | 1.17 (5.2%) | 1.23 (4.7%) | 0.91 (3.5%) |
| *GSR MFRSR/BSRN* | +0.76 (+3.2%) | 0.53 (2.7%) | 0.93 (3.8%) | 0.82 (3.4%) |



**Table 4.** Statistics for the bias between instantaneous GSR performed with PYR and MFSRS with respect to GSR BSRN (GSR XXX – GSR BSRN) at IZO between May and June 2015 (27 cloud-free days) (in $Wm^{-2}$) for different range of SZA. MB, median bias; STD, standard deviation and RMSE, root mean square error. The statistics for the relative bias are in brackets (in %).

| | Range of SZA (°) | MB | STD | RMSE |
|---|---|---|---|---|
| | **<10°** | -31.40 (-2.8%) | 10.77 (0.9%) | 33.19 (3.0%) |
| | **10-20°** | -31.03 (-2.9%) | 17.72 (1.6%) | 35.75 (3.3%) |
| **GSR PYR/BSRN** | **20-30°** | -23.27 (-2.3%) | 21.71 (2.1%) | 31.82 (3.1%) |
| | **30-40°** | -15.82 (-1.7%) | 25.32 (2.8%) | 29.85 (3.3%) |
| | **40-50°** | -11.31 (-1.4%) | 33.16 (4.2%) | 35.03 (4.5%) |
| | **50-60°** | -5.64 (-0.9%) | 46.31 (7.5%) | 46.64 (7.4%) |
| | **<10°** | -0.34 (0.1%) | 6.47 (0.6%) | 6.48 (0.6%) |
| | **10-20°** | -1.36 (-0.1%) | 8.79 (0.8%) | 8.89 (0.8%) |
| **GSR MFRSR/BSRN** | **20-30°** | -1.96 (-0.2%) | 12.17 (1.2%) | 12.33 (1.2%) |
| | **30-40°** | 0.34 (0.1%) | 14.33 (1.6%) | 14.33 (1.6%) |
| | **40-50°** | -0.35 (0.1%) | 17.70 (2.3%) | 17.70 (2.3%) |
| | **50-60°** | 0.85 (0.1%) | 19.46 (3.1%) | 19.48 (3.1%) |

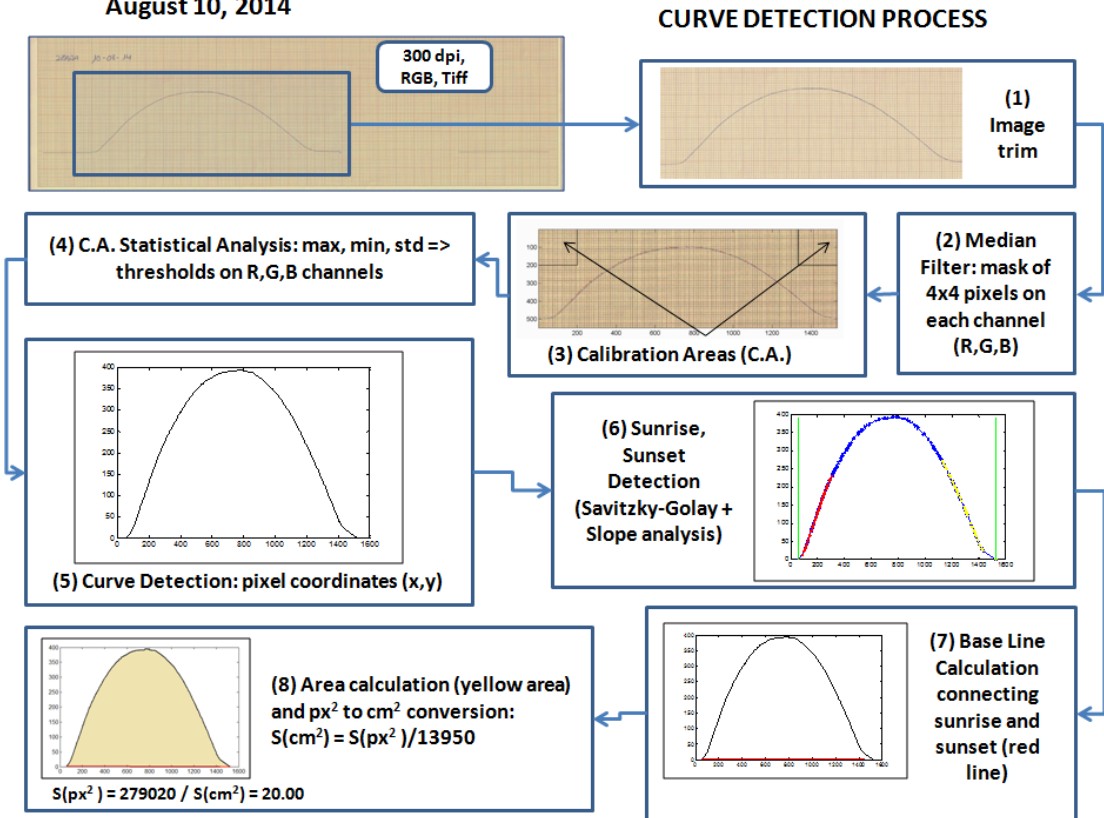

**Figure 1.** Flow chart of the method used to extract the areas from the strip cards of the pyranograph.





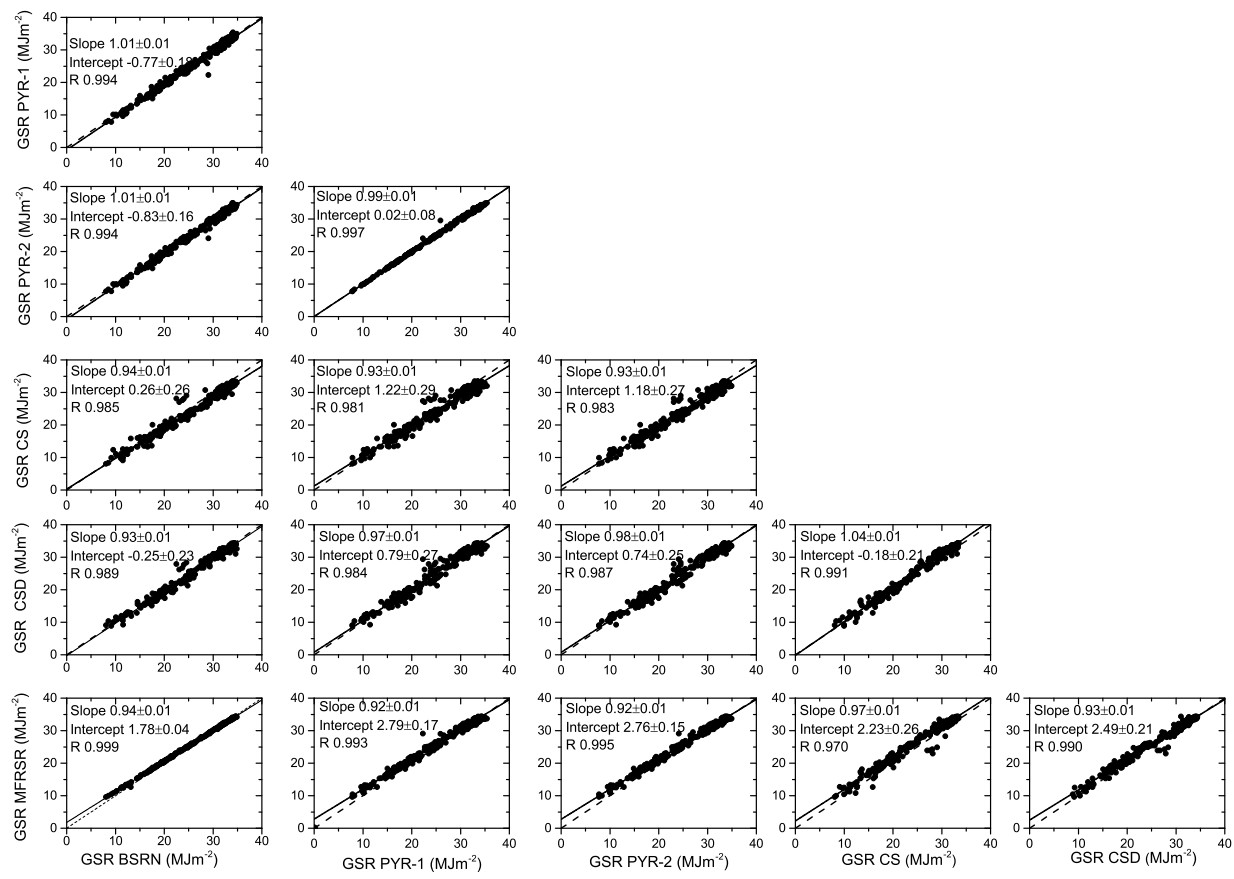

**Figure 2.** Scatterplot of daily GSR (MJm$^{-2}$) performed with CM-21 (BSRN), pyranometer (PYR) determined from Method 1 (PYR-1) and Method 2 (PYR-2), CS recorder, CSD recorder and MFRSR radiometer between July, 17 2014 and July, 12 2015 at IZO (N=272 days). The black solid lines are the least-square fits and the dotted lines are the diagonals (x=y). The least-square fit parameters are shown in the legend (slope, intercept and correlation coefficient R). N: 272 days.





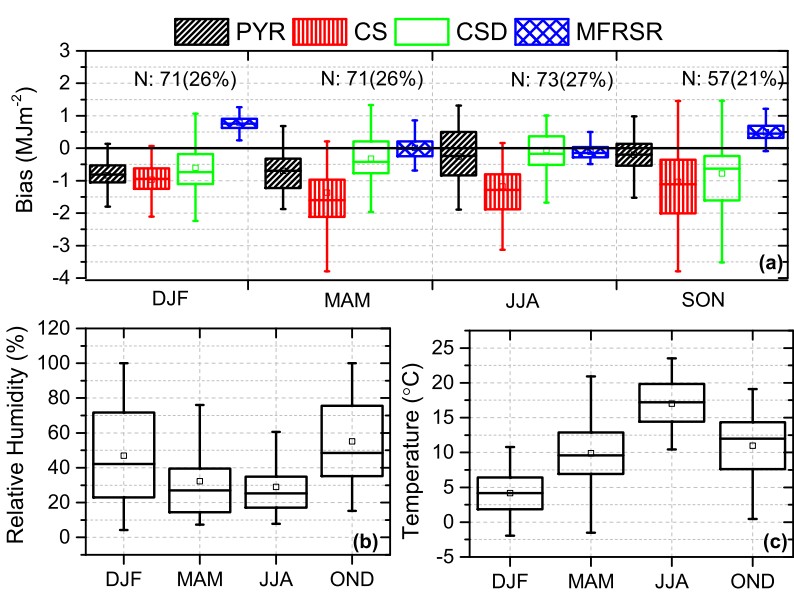

**Figure 3.** The seasonal bias: (a) GSR PYR (black), GSR CS (red), GSR CSD (green) and GSR MFRSR (blue) with respect to GSR BSRN in MJm$^{-2}$, (b) relative humidity (%) and (c) temperature (°C) between July, 17 2014 and July, 12 2015 at IZO. (DJF: December-January-February; MAM: March-April-May; JJA: June-July-August and SON: September-October-November). Lower and upper boundaries for each box are the 25th and 75th percentiles, the solid line is the median value, and hyphens are the maximum and minimum values. The numbers (N) indicate the number the measurements in each season.





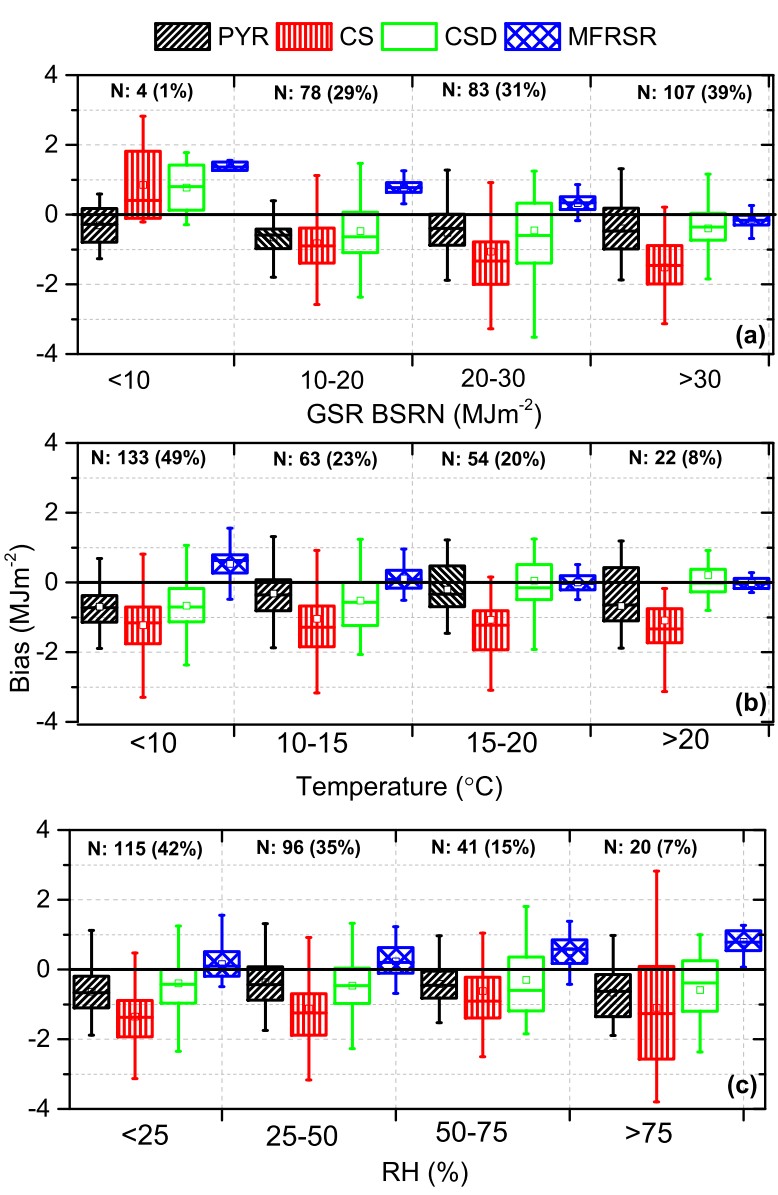

**Figure 4.** Box plot of bias (PYR: black, CS: red, CSD: green and MFRSR: blue) versus (a) corresponding GSR BSRN measurements (MJm$^{-2}$), (b) temperature in °C, and (c) relative humidity (RH) in % between July, 17 2014 and July, 12 2015 at IZO. Box plot are defined in Figure 3.





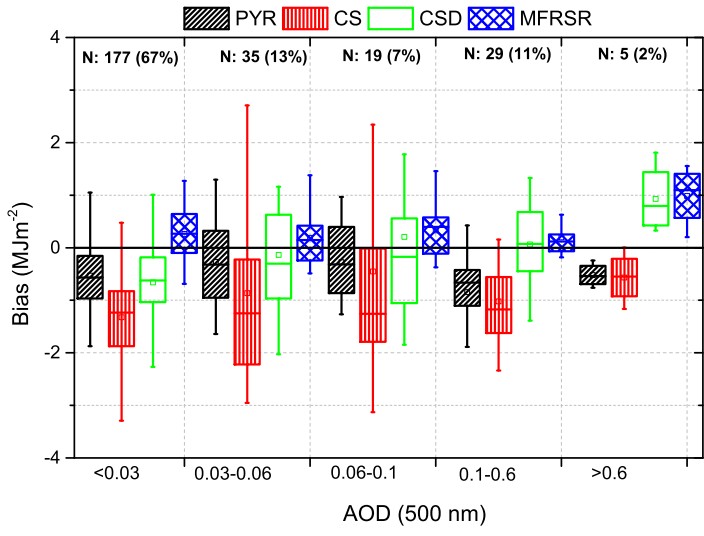

**Figure 5.** Box plot of bias (PYR: black, CS: red, CSD: green and MFRSR: blue) versus aerosol optical depth (AOD) at 500 nm between July, 17 2014 and July, 12 2015 at IZO. Box plot are defined in Figure 3.

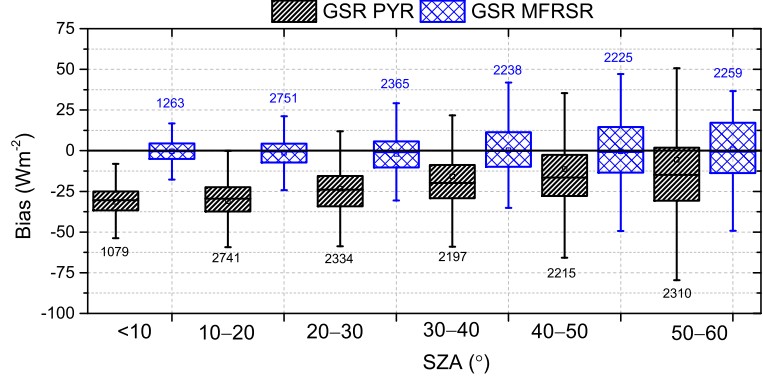

**Figure 6.** Box plot of the bias (PYR: black and MFRSR: blue) with respect to GSR BSRN in $Wm^{-2}$ at different range of SZA between May and June 2015 (27 cloud-free days) at IZO. Box plots are defined as in Figure 3.





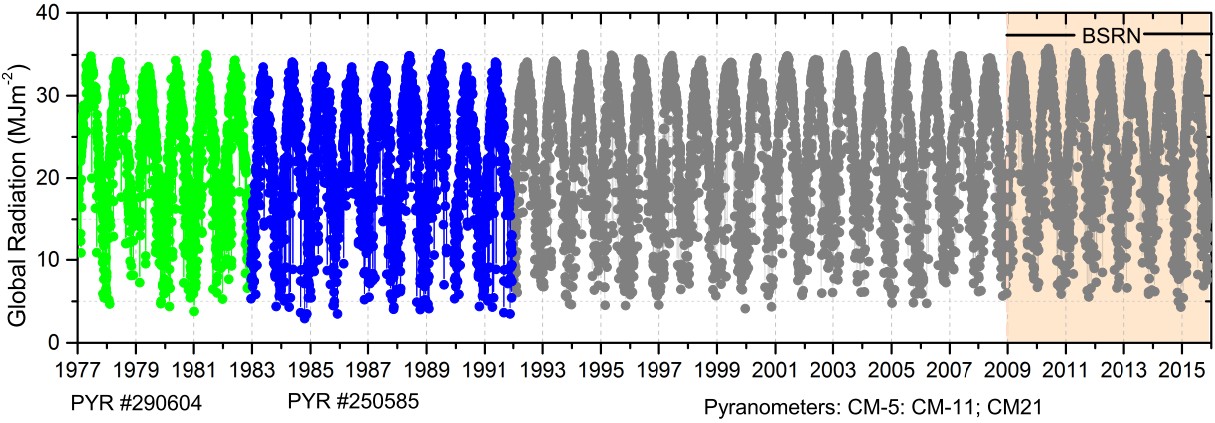

**Figure 7.** Daily GSR time series between 1977 and 2015 at IZO. The green and blue dots correspond to the measurements performed with PYR #290609 and #250585, respectively, between 1977 and 1991, and the gray dots represent the measurements performed with different pyranometers (CM-5, CM-11 and CM-21) between 1992 and 2015.

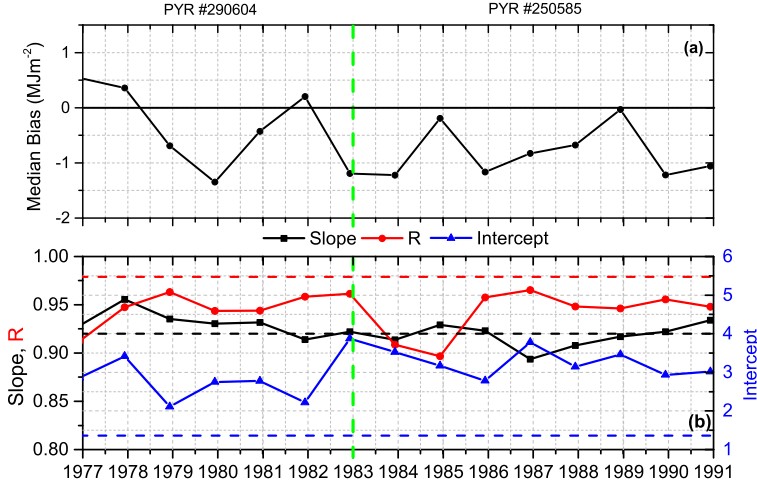

**Figure 8.** (a) Time series of the median bias between GSR PYR and GSR CS-AP in $MJm^{-2}$, and (b) time series of the slope (black line), correlation coefficient (red line) (R) and intercept (blue line) for each year. The black, red and blue dashed lines indicate the slope, R and intercept obtained in the period 2014-2015, respectively (see Figure 3). The green dashed lines represent the instrument change in 1983.