# Peer review of "Compatibility of different measurement techniques. Long-term global solar radiation observations at Izaña Observatory"

_Atmospheric Measurement Techniques, 2016_

## Referee Comment (RC1) · Anonymous Referee #3 · 28 Nov 2016

The manuscript describes the calibration and validation of different instrument types to determine solar irradiance and daily total solar radiation energy. Several of these instruments were used at Izana Observatory, when pyranometers were not yet available, and the aim of the study was to first derive their uncertainties and second to calculate the solar radiation time-series over the period 1977 to the present. While the sunshine duration instruments can only be used to derive daily totals of solar shortwave radiation, the bimetallic pyranometers provide solar irradiance levels and by numerical integration can calculate the daily totals. The final product is a time-series of daily total solar radiation energy levels for the period 1977 to the present, consisting of bimetallic pyranometers for the first part of the period and different Kipp&Zonen pyranometers for

the second part.

Even though the study has been performed with care, I think the study needs to be revised and some concerns addressed before it can be published. My main concerns are listed below:

1) The results are interesting and produce useful information with regard to the uncertainties one can expect with historical instruments when deriving daily total solar irradiation. However as pointed out by the authors themselves, such studies have been performed previously (see reference list, and especially Coulson, 1975, Garcia et al., 2014c, McArthur, 2005, Garg and Garg, 1993), so that this manuscript essentially confirms the results from these studies (see page 11, line 32) but does not add anything really novel. The authors should stress how their results differ from these previous studies.

2) The second objective of the paper was to derive a time-series of solar irradiation levels from historical measurements made at Izana Observatory. The recovered time-series were derived using the bimetallic pyranometers for the period 1977 to 1991 and Kipp&Zonen pyranometers from 1991 to the present (see figure 7). However, sunshine duration meters were deployed at Izana Observatory since 1917 (page 3, line1) and this study demonstrated that daily total irradiation levels could be derived from sunshine duration meters with comparable uncertainties to the bimetallic pyranometers as shown in Table 3. So why not extend the time-series to 1917 using these instruments?

3) I have issues with the derivation of the calibration factor F for the bimetallic pyranometers:

a) It is derived as a monthly factor using modelled solar irradiance and irradiation as reference (page 6). I do not understand why not use the pyranometer measurements from the BSRN instrument, which are certainly much more reliable than modelling results?

b) I am missing a discussion (and figure) on the amount and source of the variability of F, as I wonder why it is not be an instrument constant instead of a monthly varying factor. I would only expect it to vary slowly in time due to instrument degradation for example.

c) There seems to be a circular reasoning in the method used to derive the time-series from 1977 to 1991 (see page 10, last paragraph). The method uses modelled GSR data to retrieve the instrument calibration factors over this time-period, and then applies these calibration factors to the data to derive the solar irradiation from the instruments. But then, how can the solar irradiation levels recalculated from the bimetallic pyranometers contain any more information than the modelled radiation used in the first place to derive the calibration factors? I would suggest to carefully analyse the variability of F, and check if the calibration factors derived in 2014-2015 could not be used for the period 1977-1991.

Some more technical comments :

4) The use of the term global solar radiation (GSR) by the authors is very confusing: Sometimes it is used for the global solar irradiance (W/m2), at other times it represents daily total irradiation (J/m2). The authors should clearly distinguish between these two different parameters, and not use the same term GSR.

5) page 10, section 5: Can the neural networks track sahara dust events and volcanic eruptions (Pinatubo in 1991 and El Chichon in 1982), which are unpredicted events with significant influences on the GSR and the largest sources of aerosols at Izana observatory, for otherwise low AOD background conditions?

Abstract: Please define SD here, not later in the text.

Page 4, line 12: Please explain the method by which the pyranometer performs the diffuse and global measurements (simultaneously?).

Page 6, line 2 : As mentioned previously, why not use the BSRN Pyranometer here,

instead of the modelled solar irradiance using libradtran?

Page 10, line 22, add bimetallic, in ". . .with two pyranometers: "

References: Please remove Garcia et al., 2014a as it is only AMTD and a duplicate of Garcia et al., 2014b.

Table 1: I suggest to define the basic parameter shown as magnitude for MFRSR and CM21 as solar irradiance (W/m2), from which solar irradiation (J/m2) can be derived.

Figure 3: There is a mistake in the bottom figures : OND should probably be corrected to SON?

---

## Referee Comment (RC2) · Anonymous Referee #2 · 6 Dec 2016

The study presents an inter-comparison of different measurement techniques of global solar radiation. The authors demonstrate consistency and analyze uncertainties between the measurements by old and modern instruments using one year (July 2014 to July 2015) of simultaneous observations at the high altitude Izana Atmospheric Observatory, Tenerife, Spain. The uncertainties were analyzed as a function of different seasons, intensity of solar irradiance, ambient temperature, relative humidity (RH), aerosol optical depth (AOD), and solar zenith angle (SZA). The obtained results are then applied for validation of a long-term data series started in 1977. This is an interesting study on compatibility of different measurement techniques that also has an application of extending the global solar radiation time series. However, while graphical

presentation of the results is quite clear, I found that the text of the manuscript might be improved prior the possible publication in AMT. In particular the "Introduction" and "Results" sections need more work. I think that the authors could spend more efforts on revision of the text, clarity and completeness of the presentation. Below please find my general and specific comments.

1. The authors claim to analyze performance of the measurements as a function of AOD, but I think that observations in this high altitude (∼2400 m a.s.l.) atmospheric observatory does not appropriate for this type of analysis. The range of the AOD variability is very small and the site is characterized by very low atmospheric aerosol loading. In fact, 67 % of the observations are for AOD(500nm) < 0.03. Only 5 observations correspond to AOD > 0.6, and I suspect these observations are from a couple of consequent days and belonging to the same aerosol event (probably a case of dust transport). Therefore, the conclusions derived regarding compatibility of different measurement techniques under varying aerosol conditions maybe not solid enough.

2. Inter-comparison of the measurement techniques is also presented as a function of seasons. What seasonal characteristic is expected to influence performance of the instruments? The inter-comparison is then presented as a function of temperature. I have strong impression that the results for summer months are similar to the results for temperature range of 15-20 and >20 degree; same for winter and <10 degree. I think that the analysis vs. seasons is redundant with the analysis vs. temperature.

3. p9, line 9 it is written: "underestimated for all sensors and instruments except for GSR MFRSR in winter and autumn". Why it could be, what is special in this case? An explanation/hypothesis?

4. The introduction section could include a review on the previous inter-comparison and compatibility works. What is similar or different/new in the suggested here study? Several references are mentioned in "Summary and conclusions" section reporting 10 and 20 % uncertainties obtained in previous studies. The authors report much lower

uncertainties. Is it due to different measurement conditions, quality of the reference instrument? A discussion or analysis can be useful.

5. It is mentioned in the abstract that fraction of the clear sky is among the factors that were found to affect the global solar radiation. It is mention among such factors as temperature, RH, and SZA. While I see analysis vs. temperature, RH and SZA, it is not clear for me what the authors mean by fraction of the clear sky and where it is in the manuscript.

6. Specific comments: p1, line 12: "By comparing with . . ." the sentence is not clear. P2, line1: Should not the reference to [Stanhill and Cohen 2001] be cited separately from others and just after word "dimming"? Please check. p8, line 8: "solar irradiation" to "solar irradiance" ? p8, line 12: "July, 17 2014" to "July 17, 2014" etc. p9, line 12: "There is not dependence . . ." to "There is no dependence . . ." p9, line 20: "It is clear a dependence with the irradiance level, the larger BSRN GSR values the larger bias." please revise the sentence construction. p9, line 21: "A slightly dependence with. . ." to "A slight dependence with. . ." Section 5, in the beginning: It can facilitate the reading if some principles of the applied artificial neural networks will be shortly described in the text, instead to address the readers to the bibliography. p. 11, line 18: I think that the next sentence can be reformulated, also "goodness" does not sound in this context. "However, the intercept is significantly higher in the 1977-1991 period than in the 2014-2015 period that might be likely due to, after reinstallation a correction in the bias, to instrument cleaning and fit. The goodness of the fit is noticeable during the whole period. . ."

Finally, this is only a suggestion, but the authors may reconsider the title and indicate already in the first sentence that it is about measurements techniques of global solar radiation, otherwise "Compatibility of different measurement techniques." sounds too general. For example, "Compatibility of different measurement techniques of global solar radiation and application for long-term observations at Izaña Observatory"

---

## Author Comment (AC1) · 16 Jan 2017

**amt-2016-293**

Interative comment on "Compatibility of different measurement techniques. Long-term global solar radiation observations at Izaña Observatory" by R. D. García et al.

**Anonymous Referee #3:**

The manuscript describes the calibration and validation of different instrument types to determine solar irradiance and daily total solar radiation energy. Several of these instruments were used at Izaña Observatory, when pyranometers were not yet available, and the aim of the study was to first derive their uncertainties and second to calculate the solar radiation time-series over the period 1977 to the present. While the sunshine duration instruments can only be used to derive daily totals of solar shortwave radiation, the bimetallic pyranometers provide solar irradiance levels and by numerical integration can calculate the daily totals. The final product is a time-series of daily total solar radiation energy levels for the period 1977 to the present, consisting of bimetallic pyranometers for the first part of the period and different Kipp & Zonen pyranometers for the second part.

Even though the study has been performed with care, I think the study needs to be revised and some concerns addressed before it can be published. My main concerns are listed below:

**Authors: We appreciate the positive and constructive comments of the Referee. Below we discuss and respond to his/her general and specific comments.**

 The results are interesting and produce useful information with regard to the uncertainties one can expect with historical instruments when deriving daily total solar irradiation. However as pointed out by the authors themselves, such studies have been performed previously (see reference list, and especially Coulson, 1975, Garcia et al., 2014c, McArthur, 2005, Garg and Garg, 1993), so that this manuscript essentially confirms the results from these studies (see page 11, line 32) but does not add anything really novel. The authors should stress how their results differ from these previous studies.

**Authors: This work presents two fundamental novelties respect to previous studies (i.e. Coulson, 1975; García et al., 2014c; McArthur, 2005; Garg and Garg, 1993):**

- 1. It has been possible to assess and verify the results obtained with different instrumentation by the authors mentioned above in a testbed station (Izaña station) by performing simultaneous measurements under the same environmental conditions. Moreover, our study is based on measurement comparison of an annual cycle using classical and modern radiation and sunshine duration instruments.
- 2. An important characteristic of the Izaña station is the wide range of variability found in some key atmospheric parameters and meteorological variables

throughout the year, i.e., aerosol optical depth, from Rayleigh conditions (< 0.02) to dusty conditions (>0.5), Ångström exponent from >1 to ~0, relative humidity from <5% to 100%, etc... Furthermore, as a subtropical region site the sun reaches very low SZA (5-6°) and provides very high radiation dosis.

3. The methodology and the results obtained in this work might be applicable to any other station.

The authors have added this information in the section: Summary and Conclusions of the final manuscript as follows:

"Assuming GSRH from BSRN as reference, the measured or estimated GSRH values show median biases of 2% and 1% for PYR GSRH and MFRSR, respectively, and of 5% and 2% for CS GSRH and CSD GSRH, respectively. These results, as expected, show that the instruments that measure directly GSRH, such as the PYR and MFRSR, present lower MB and lower scatter than the ones that estimate the GSRH, such as the CS and CSD recorders. Moreover, median bias values for each instrument are within their corresponding uncertainty, agreeing with results obtained by other authors (Coulson, 1975; García et al., 2014c; McArthur, 2005). The comparison of the daily GSRH values from PYR and MFRSR showed a good agreement with GSRH BSRN, obtaining a RMSE of 0.9 ~MJm-2 (3%) and ~0.5 MJm-2 (2%) for PYR GSRH and MFRSR GSRH, respectively, and ~1.7 MJm-2 (7%) and ~1.1 MJm-2 (4%) for CS GSRH and CSD GSRH, respectively. It is worth highlighting the fact that the biases for PYR found in this study are lower than those reported by others authors. For example, Coulson (1975); Garg and Garg (1993) obtained uncertainties between 10 and 20% reduced up to 4-5% by Esteves and de Rosa (1989) and Soulayman and Daudé (1995). These results, obtained with simultaneous observations under the same environmental conditions, provide information about expected GSRH uncertainties from historical instruments useful for assessing long-term GSRH data series constructed from classical and modern instruments. "

2. The second objective of the paper was to derive a time-series of solar irradiation levels from historical measurements made at Izaña Observatory. The recovered time series were derived using the bimetallic pyranometers for the period 1977 to 1991 and Kipp & Zonen pyranometers from 1991 to the present (see figure 7). However, sunshine duration meters were deployed at Izaña Observatory since 1917 (page 3, line1) and this study demonstrated that daily total irradiation levels could be derived from sunshine duration meters with comparable uncertainties to the bimetallic pyranometers as shown in Table 3. So why not extend the time-series to 1917 using these instruments?

Authors: One of the main objectives of this paper was to present a GSR data time series at Izaña station only from GSR instruments. From 1917 to 1976 the GSR is obtained by estimations from sunshine duration measurements. The long term GSR data from both measurements and estimations will be the subject of a future analysis.

**3.** I have issues with the derivation of the calibration factor F for the bimetallic pyranometers:

a) It is derived as a monthly factor using modelled solar irradiance and irradiation as reference (page 6). I do not understand why not use the pyranometer measurements

from the BSRN instrument, which are certainly much more reliable than modeling results?

Authors: Garcia et al. (2014c) demonstrate that LibRadtran model to be a very useful tool as a measurement quality control, obtaining that there is a good agreement between measurements and simulations with the mean bias (simulations-measurements) of -0.30 MJm-2 (-1.1%) and root mean square error of 0.38 MJm-2 (1.3%) for global solar radiation for 386 days between March 2009 and August 2012. Based on these results, we determine the coefficients  $F_m$  from the simulated measurements. On the other hand the reference to validate the method is the BSRN GSR and therefore we do not use it to calculate the coefficients  $F_m$ .

Any way, following the referee's recommendation, we have determined the coefficients  $F_m$  from the BSRN pyranometer measurements and compared with those obtained with the LibRadtran model simulations. The results obtained are quite similar in both cases (Figure 1) with root mean square error of 0.54 MJm-2(1.15%) (Figure 1b). Thus both coefficients could be used indifferently.

Figure 1.- (a) Annual cycle of Fm coefficients determined from the BSRN pyranometer measurements (black bars) and determined from the LibRadtran model simulations (gray bars). (b) Scatterplot of daily GSR obtained from LibRadtran model simulations versus daily GSR obtained from BSRN pyranometer measurements. The fit coefficients and Pearson correlation coefficient are shown in the legend.

b) I am missing a discussion (and figure) on the amount and source of the variability of F, as I wonder why it is not be an instrument constant instead of a monthly varying factor. I would only expect it to vary slowly in time due to instrument degradation for example.

Authors: Figure 1a shows a clear annual cycle of  $F_m$ , with lower values in winter and higher in summer. Therefore the use, for example, of an annual mean value of  $F_m$  to recover the pyranograph measurements would lead to an important source of error. If we calculate the  $F_m$  in the period 1977-1991 by using the model simulations with the

adequate input variables, the obtained GSR estimations have accuracy within the instrumental error as demonstrated in Figure 1b, and thus we obtain the  $F_m$  series showed in Figure 2. In general, every year a similar behavior to that of Figure 1a is observed, but not always, in 1979-1981, 1986, 1987  $F_m$  differs from the annual cycle showed in Figure 1a. Moreover, there is a significant difference in the values obtained before and after 1983, when an instrument change took place.

Figure 2.- Time series  $F_m$  (MJ/m2/cm2) in the period 1977-1991 and reference indicates the period 2014-2015.

Many works pointed that the main source of variability in the bimetallic pyranograph measurements is its strong temperature dependence and its degradation with time (Coulson, 1975; Garg and Garg, 1993; Esteves and de Rosa, 1989). From Figure 3 we observe there is no dependence of S, and  $H_{ci}$  on temperature (Figures 3a and 3b) for the reference period (2014-2015). However there is a clear dependence of the ( $H_{ci}$  / S) ratio with the temperature (Figure 3c) in this period, explaining the annual cycle.

---

## Author Comment (AC2) · 16 Jan 2017

The study presents an inter-comparison of different measurement techniques of global solar radiation. The authors demonstrate consistency and analyze uncertainties between the measurements by old and modern instruments using one year (July 2014 to July 2015) of simultaneous observations at the high altitude Izaña Atmospheric Observatory, Tenerife, Spain. The uncertainties were analyzed as a function of different seasons, intensity of solar irradiance, ambient temperature, relative humidity (RH), aerosol optical depth (AOD), and solar zenith angle (SZA). The obtained results are then applied for validation of a long-term data series started in 1977. This is an interesting study on compatibility of different measurement techniques that also has an application of extending the global solar radiation time series. However, while graphical presentation of the results is quite clear, I found that the text of the manuscript might be improved prior the possible publication in AMT. In particular the "Introduction" and "Results" sections need more work. I think that the authors could spend more efforts on revision of the text, clarity and completeness of the presentation. Below please find my general and specific comments.

*Authors: We acknowledge the referee's constructive comments. The main text of the manuscript, and specifically "Introduction" and "Results" sections, have been reviewed and smoothed, and in the following we discuss and respond to the general and specific comments.*

1. The authors claim to analyze performance of the measurements as a function of AOD, but I think that observations in this high altitude (2400 m a.s.l.) atmospheric observatory does not appropriate for this type of analysis. The range of the AOD variability is very small and the site is characterized by very low atmospheric aerosol loading. In fact, 67 % of the observations are for AOD (500nm) < 0.03. Only observations correspond to AOD > 0.6, and I suspect these observations are from a couple of consequent days and belonging to the same aerosol event (probably a case of dust transport). Therefore, the conclusions derived regarding compatibility of different measurement techniques under varying aerosol conditions maybe not solid enough.

   *Authors: We agree. Following the Referee's recommendation, the authors have removed this analysis given that the range of the AOD variability is rather small (i.e. only 2% (N: 5 days) of the days present AOD>0.6). Accordingly the Figure 5 has been removed.*

*However the authors consider interesting to include a short assessment on the bias found among BSRN GSR and the rest of measurements (PYR, CS, CSD and MFRSR) considering both background conditions (AOD<0.10) and dust conditions (AOD≥0.10) based on García et al. (2014c). This analysis was added in the manuscript as follows:*

*"As aforementioned, some of the analysis instruments and methods are sensitive to different factors and atmospheric conditions. We have analyze GSR the differences with respect to the BSRN GSR$_H$ in function of the solar irradiance (Figure 3a), the average temperature and relative humidity (RH) (Figure 3b and 3c, respectively), the FCS, and AOD."*

*"We have also studied the differences with respect to FCS and AOD (not shown here). No dependence with FCS was found, although it should be noted that 85% of the days (N: 232 days) presented FCS>75% while only 1% (N: 4 days) showed FCS<25%. Concerning the dependence on AOD only background conditions (AOD<0.10) and dust conditions (AOD≥0.10) have been considered based on García et al. (2014b). No dependence on AOD is found, although we must highlight the fact that 87% of the days (N: 231 days) presented AOD<0.10 and 13% of the days (N: 33 days) showed AOD≥0.10. The GSR$_H$ measurements most affected by AOD were those obtained with the CSD, which shows a monotonic dependence in the bias, being negative for pristine skies and positive for dust conditions."*

2. **Inter-comparison of the measurement techniques is also presented as a function of seasons. What seasonal characteristic is expected to influence performance of the instruments? The inter-comparison is then presented as a function of temperature. I have strong impression that the results for summer months are similar to the results for temperature range of 15-20 and >20 degree; same for winter and <10 degree. I think that the analysis vs. seasons is redundant with the analysis vs. temperature.**

*Authors: The referee is indeed right. So, the authors have decided to remove the inter-comparison presented as a function of seasons, given that this information is redundant with that provided by the inter-comparison as a function of temperature. Corresponding text to these results in Section 4.1 has been rewritten.*

3. **p9, line 9 it is written: "underestimated for all sensors and instruments except for GSR MFRSR in winter and autumn". Why it could be, what is special in this case? An explanation/hypothesis?**

*Authors: MFRSR presents a dependence with respect to the temperature and relative humidity. As remarked before the analysis respect to the season is identical to the analysis with respect to the temperature. This sentence has been rephrased as follows: "…underestimated for all sensors and instruments except for MFRSR GSR for low temperature values". The MFRSR is thermally controlled in order to operate at its optimal working-temperature, that is, 40°C. Thus, if the temperature is below this value the instrument head is heated until the working-temperature is reached, and then the heating is switched off. Therefore, when the difference between ambient temperature and 40 ° C is very large, the heating system is continuously working at 100% effort, creating in this former version of MFRSR instrument an electromagnetic*

*frequency interference with the irradiance measurements leading to inaccurate measurements (Harrison, 1994; Hodges and Michalsky, 2011).*

*The authors have added this information in the final manuscript as follows:*

*"Finally the MFRSR is the instrument that shows the best performance, with a bias close to zero through the whole year, and the lowest scatter (Figure 3). We observe, in general, an overestimation in MFRSR $GSR_H$, unlike the rest of the compared instruments. The MFRSR $GSR_H$ has a clear positive dependence with irradiance (Figure 3a). There is not temperature dependence for temperatures >15°C, (Figure 3c) and a slight dependence, for lower temperatures. The MFRSR is thermally controlled at around 40°C. Thus, when the difference between ambient temperature and 40°C is very large, the heating system is continuously working at 100% effort, creating in the former versions of MFRSR instrument an electromagnetic frequency interference with the irradiance measurements, leading to higher measurement inaccuracies (Harrison, 1994; Hodges and Michalsky, 2011). The MFRSR $GSR_H$ measurements show a slight positive dependence with relative humidity (Figure 3c)."*

4. **The introduction section could include a review on the previous inter-comparison and compatibility works. What is similar or different/new in the suggested here study? Several references are mentioned in "Summary and conclusions" section reporting and 20 % uncertainties obtained in previous studies. The authors report much lower uncertainties. Is it due to different measurement conditions, quality of the reference instrument? A discussion or analysis can be useful.**

*Authors: Following the Referee's recommendations, the authors have added the following information in the Introduction:*

*"In order to complete and extend the $GSR_H$ time series, ancillary measurements are often used to estimate the $GSR_H$. However, it is necessary to know the accuracy than estimations by comparison with $GSR_H$ measurements simultaneity performed with modern instruments. The sunshine duration (SD) has been widely used by applying the well-known Ångström-Prescott equation (Angstrom, 1924, 1956; Prescott, 1940) to estimate the $GSR_H$. Several authors as Almorox et al. (2005), Yorukoglu and Celik (2006) and García et al. (2014c) have used this method in different regions obtaining very similar results. Regarding the bimetallic pyranometer (PYR), designed in the early 1920s (Robitzsch, 1926) and widely used until the late 1960s, the $GSR_H$ is obtained from an equation involving the recorded area and the ambient temperature (Robitzsch, 32). Stravisi (1986) performed a posteriori calibration of a PYR over a 3-year period obtaining hourly, daily and monthly correction factors. Later, Esteves and de Rosa (1989) proposed a correction method to improve the accuracy of daily averaged $GSR_H$ readings, reducing the error from 20% to ~4%. Maxwell et al. (1999) performed a comparison between $GSR_H$ estimations from a PYR and $GSR_H$ measurements with an Eppley PSP radiometer. They applied an automatized process to scan the PYR charts finding differences in daily $GSR_H$ values ranging between 2% and 10% over the course of a year."*

*However, all these partial intercomparisons were performed in several sites with different environmental conditions, and different instruments and time periods. On the contrary, what we propose in this study is to know the performance of different instruments running in parallel in a testbed site where the environmental conditions show a wide range of variation throughout the year. This allows us to obtain a comprehensive and consistent assessments on the $GSR_H$ differences obtained with these instruments."*

5. **It is mentioned in the abstract that fraction of the clear sky is among the factors that were found to affect the global solar radiation. It is mention among such factors as temperature, RH, and SZA. While I see analysis vs. temperature, RH and SZA, it is not**

**clear for me what the authors mean by fraction of the clear sky and where it is in the manuscript.**

*Authors: There was an error in the abstract and it should say* "…*Factors such as temperature, relative humidity (RH) or the solar zenith angle (SZA) have shown to moderately affect the GSR$_H$ observations…*"

*The authors have defined the fraction of clear sky (FCS) in Equation 6 (p7) of the manuscript as the ratio between the maximum daily sunshine duration and sunshine duration performed with CS recorder (García et al. (2014 a,b)). No significant bias between GSR BSRN and the rest of measurements (PYR, CS, CSD and MFRSR) respect to the fraction of clear sky (FCS) has been found despite 85% of the days in the year intercomparison (N: 232 days; 85%) showed FCS>75%*

[Figure]

*Figure 1.- Box plot of bias (PYR: black; CS: red; CSD: green and MFRSR: blue) versus fraction clear sky (FCS) in % between July 17, 2014 and July 12, 2015 at IZO.*

**Specific comments:**

**p1, line 12: "By comparing with: " the sentence is not clear.**

*Authors: The sentence has been modified as follows:*

"…*As an application of the methodology developed in this work, we have re-evaluated the GSR$_H$ time series performed at IZO with two PYRs between 1977 and 1991. Their high consistency and temporal stability have been stated by comparing with GSR$_H$ estimates obtained from SD observations…*"

**P2, line1: Should not the reference to [Stanhill and Cohen 2001] be cited separately from others and just after word "dimming"?**
*Authors: Done*

**Please check. p8, line 8: "solar irradiation" to "solar irradiance"?**
*Authors: Done*

**p8, line 12: "July, 17 2014" to "July 17, 2014" etc.**
*Authors: Done*

**p9, line 12: "There is not dependence: "to "There is no dependence…"**
*Authors: Done*

**p9, line 20: "It is clear a dependence with the irradiance level, the larger BSRN GSR values the larger bias." please revise the sentence construction.**

*Authors: The sentence has been modified as follows:*

*"The CS $GSR_H$ estimations present a clear dependence with the irradiance, with higher bias for higher BSRN $GSR_H$ values."*

**p9, line 21: "A slightly dependence with:" to "A slight dependence with:**
*Authors: Done*

**Section 5, in the beginning: It can facilitate the reading if some principles of the applied artificial neural networks will be shortly described in the text, instead to address the readers to the bibliography.**

*Authors: The artificial neural networks methodology was already published by García et al. (2016), and it is not specifically used in this study.*

**p. 11, line 18: I think that the next sentence can be reformulated, also "goodness" does not sound in this context. "However, the intercept is significantly higher in the 1977-1991 period than in the 2014-2015 period that might be likely due to, after reinstallation a correction in the bias, to instrument cleaning and fit. The goodness of the fit is noticeable during the whole period: "**

*Authors: The sentence has been modified as follows:*

*"…The intercept in the period 1977-1991 is higher than in the 2014-2015 period, while the R values are of about 0.95 in the whole period, and 0.98 in the 2014-2015 period. This improvement is likely due to the cleaning and fitting of the instrument before being re-installed for the 2014-2015 inter-comparison."*

**Finally, this is only a suggestion, but the authors may reconsider the title and indicate already in the first sentence that it is about measurements techniques of global solar radiation, otherwise "Compatibility of different measurement techniques." sounds too general. For example, "Compatibility of different measurement techniques of global solar radiation and application for long-term observations at Izaña Observatory"**

*Authors: The tittle proposed by the Referee has been accepted.*